# *NGT1* Is Essential for *N*-Acetylglucosamine-Mediated Filamentous Growth Inhibition and *HXK1* Functions as a Positive Regulator of Filamentous Growth in *Candida tropicalis*

**DOI:** 10.3390/ijms21114036

**Published:** 2020-06-05

**Authors:** Qiuyu Zhang, Li Xu, Sheng Yuan, Qinghua Zhou, Xuxia Wang, Lei Wang, Zhiming Hu, Yunjun Yan

**Affiliations:** Key Laboratory of Molecular Biophysics, the Ministry of Education, College of Life Science and Technology, Huazhong University of Science and Technology, Wuhan 430074, China; zhangqiuyu@hust.edu.cn (Q.Z.); xuli@mail.hust.edu.cn (L.X.); m201871755@hust.edu.cn (S.Y.); qinghuazhou@hust.edu.cn (Q.Z.); 2017506005@hust.edu.cn (X.W.); 2017511004@hust.edu.cn (L.W.); D201577414@hust.edu.cn (Z.H.)

**Keywords:** *Candida tropicalis*, *N*-acetylglucosamine, *NGT1*, *HXK1*, filamentous growth, cell growth

## Abstract

*Candida tropicalis* is a pathogenic fungus that can cause opportunistic infections in humans. The ability of *Candida* species to transition between yeast and filamentous growth forms is essential to their ability to undergo environmental adaptation and to maintain virulence. In other fungal species, such as *Candida albicans*, *N*-acetylglucosamine (GlcNAc) can induce filamentous growth, whereas it suppresses such growth in *C. tropicalis*. In the present study, we found that knocking out the GlcNA-specific transporter gene *NGT1* was sufficient to enhance *C. tropicalis* filamentous growth on Lee’s plus GlcNAc medium. This suggests that GlcNAc uptake into *C. tropicalis* cells is essential to the disruption of mycelial growth. As such, we further studied how GlcNAc catabolism-related genes were able to influence *C. tropicalis* filamentation. We found that *HXK1* overexpression drove filamentous growth on Lee’s media containing glucose and GlcNAc, whereas the deletion of the same gene disrupted this filamentous growth. Interestingly, the deletion of the *DAC1* or *NAG1* genes impaired *C. tropicalis* growth on Lee’s plus GlcNAc plates. Overall, these results indicate that *HXK1* can serve as a positive regulator of filamentous growth, with excess GlcNAc-6-PO_4_ accumulation being toxic to *C. tropicalis*. These findings may highlight novel therapeutic targets worthy of future investigation.

## 1. Introduction

*Candida tropicalis* is a fungal pathogen that can cause opportunistic infections in humans [1,2]. *Candida* can cause both relatively superficial infections of the oral mucosa and extremely severe systemic infections that are, in some cases, fatal [3,4] and damaging to tissues, including the spleen, liver, heart, lungs, kidneys, bone, and ocular mucosa [5,6,7,8]. Rising numbers of patients who are immunocompromised due to radiotherapy, chemotherapy, or organ transplantation have led to an increase in the incidence of opportunistic *Candida* infections [9,10]. While *C. albicans* is the most common cause of such fungal infections, *C. tropicalis* infection rates have been rapidly increasing among immunocompromised individuals, and these infections are associated with higher mortality rates [9,10,11]. In addition, *C. tropicalis* is more resistant to antifungal agents, such as fluconazole, when compared with *C. albicans*, with 15% of *C. tropicalis* isolates being fluconazole-resistant and just 3.8% *C. albicans* isolates being fluconazole-resistant [12,13]. This suggests that *C. tropicalis* is an emerging pathogen worthy of closer study and medical scrutiny.

One important determinant of fungal infectivity in mammals is the yeast-to-filament transition, as well as the ability of these fungi to adapt to changes in the local environment [14,15,16,17]. Many environmental factors, including serum conditions, elevated local CO_2_ levels, a higher temperature within hosts (37 °C), and *N*-acetylglucosamine (GlcNAc), can drive *C. albicans* filamentation [15,18,19,20]. *C. tropicalis* is evolutionarily similar to *C. albicans* [13], both of which belong to the CTG clade. A distinguishing characteristic of the species in this clade is that the CTG codon is translated into serine instead of leucine [21,22,23]. Changes in environmental conditions can also influence *C. tropicalis* filamentation, with shifts in nutrient levels, temperature, pH, and available carbon sources all driving this transition [24,25]. Yeast and filamentous forms of these *Candida* species exhibit clear differences in their morphology, with changes in cell and colony shape, and in their ability to grow invasively within the host environment [3,26]. The yeast-to-filament transition can facilitate endothelial cell penetration by these fungi and spreading within the bloodstream, often resulting in fatal systemic candidiasis [27].

In most mammals, *C. albicans* is present as a commensal microbe [28,29], whereas *C. tropicalis* is found in certain mammals and in mangrove sediments, seawater, beaches, sea sediments, and soils in tropical and subtropical regions [30,31]. As a key component of chitin and of the extracellular matrix of animals, GlcNAc is widely present in the environment and in the mammalian hosts, including in the gastrointestinal tract and in the extracellular matrix environment, where it is present within glycosaminoglycans [32,33]. Environmental GlcNAc detection is a key function of many fungal species, given that GlcNAc can be used as both a carbon source [34] and as a signaling molecule that can induce morphological changes [35,36,37].

GlcNAc can readily drive the filamentation of a range of dimorphic fungi, including *Histoplasma capsulatum*, *C. albicans*, *Yarrowia lipolytica*, and *Blastomyces dermatitidis* [36,37,38]. However, in *C. tropicalis*, GlcNAc strongly inhibits filamentous growth [39]. The Ras1-cAMP signaling pathway and many transcription factors have been shown to be involved in both the GlcNAc-induced filamentous growth of *C. albicans* and the GlcNAc-mediated inhibition of *C. tropicalis* filamentous growth [39,40]. However, *C. tropicalis* also expresses the genes needed for mediating GlcNAc transportation and catabolism, as does *C. albicans* [21,39]. There is little research on the filamentous growth regulatory mechanisms of *C. tropicalis*. The role of the GlcNAc-specific transporter gene *NGT1*, as well as the genes responsible for GlcNAc catabolism in this filamentous growth, are still unclear. The relationship between the ability of GlcNAc to inhibit filamentous growth and its metabolic processing is unclear. In the present study, we therefore sought to assess the roles of both GlcNAc catabolic genes and *NGT1* in *C. tropicalis*. We ultimately found that *NGT1* was essential for the ability of GlcNAc to inhibit filamentous growth. We additionally investigated the role of the GlcNAc catabolic genes Glucosamine-6-phosphate deaminase (*NAG1*), *N*-acetylglucosamine-6-phosphate deacetylase (*DAC1*), and GlcNAc kinase gene (*HXK1*), in regulating *C. tropicalis* filamentous growth.

## 2. Results

### 2.1. NGT1 Deletion Enhances C. tropicalis Filamentous Growth on GlcNAc-Containing Medium

Previous work has shown that, whereas GlcNAc enhances *C. albicans* filamentous growth, it has the opposite effect on *C. tropicalis* [39]. In both of these species, the *NGT1* protein is composed of several transmembrane regions [21,41] and is essential for efficient GlcNAc uptake [41]. We therefore sought to assess whether *NGT1* is similarly required in order for GlcNAc to suppress the filamentous growth of *C. tropicalis* by plating both WT and *ngt1*Δ/Δ cells onto Lee’s agar plates containing either GlcNAc or glucose at 25 °C or 37 °C for 6 days, after which the morphology of the cells and colonies was assessed. On glucose-containing plates, the filamentous growth of *ngt1*Δ/Δ, WT, and *ngt1*Δ/Δ+*NGT1p*-*NGT1* reconstituted strain cells showed no apparent difference at both tested temperatures, while the *ngt1*Δ/Δ mutant cells underwent filamentous growth more readily than the WT and *ngt1*Δ/Δ+*NGT1p*-*NGT1* reconstituted strain cells at both tested temperatures on GlcNAc-containing plates (Figure 1). We further analyzed the role of *NGT1* in the GlcNAc-mediated inhibition of *C. tropicalis* filamentous growth by overexpressing *NGT1*. We found that such overexpression had no obvious impact on filamentous growth on these two different media types (Lee’s plus glucose and Lee’s plus GlcNAc) at both 25 °C and 37 °C (Figure 1).

### 2.2. GlcNAc Induces HXK1, DAC1, NAG1, and NGT1 Expression in C. tropicalis

*NGT1* encodes a GlcNAc-specific transporter that is required for GlcNAc internalization by *C. albicans* [41], while *HXK1*, *NAG1*, and *DAC1* encode GlcNAc kinase, glucosamine-6-phosphate deaminase, and GlcNAc-6-phosphate deacetylase, respectively [21,42,43]. We next assessed how the expression of these GlcNAc catabolism-related genes was affected by GlcNAc in *C. tropicalis*. Using qRT-PCR, it was revealed that the expression of all four assessed genes (*HXK1*, *DAC1*, *NAG1*, and *NGT1*) was significantly enhanced on GlcNAc-containing media relative to glucose-containing media, with the relative induction of *NGT1* being particularly pronounced with a 120-fold increase in its expression (Figure 2). This indicated that GlcNAc was able to readily induce *HXK1*, *DAC1*, *NAG1*, and *NGT1* expression in *C. tropicalis*.

### 2.3. HXK1 Overexpression Promotes C. tropicalis Filamentous Growth

*HXK1*, *DAC1*, and *NAG1* are all needed to facilitate the catabolism of GlcNAc. The *HXK1* protein phosphorylates GlcNAc, after which *DAC1* and *NAG1* deacetylate and deaminate it, respectively, yielding fructose-6-PO_4_ [44]. We next sought to explore the role of these catabolic genes in the filamentous growth of *C. tropicalis* via individually overexpressing each of them under the control of the *C. tropicalis* TDH3 promoter. These overexpression mutants were then grown on glucose- or GlcNAc-containing media at 25 °C and 37 °C. Relative to vector control cells, those cells overexpressing *HXK1* exhibited more pronounced filamentous growth on glucose-containing media at 25 °C as well as on GlcNAc-containing media at both 25 °C and 37 °C, with the most pronounced effect being evident on GlcNAc-containing media at 25 °C (Figure 3), as vector control cells exhibited no filamentous growth under these conditions. When grown on glucose-containing Lee’s medium at 37 °C, both the WT vector and *HXK1*-overexpressing strains exhibited robust filamentous growth (Figure 3), thus masking any potential enhancement of filamentous growth under these conditions. These results indicating that overexpression *HXK1* promoted filamentous growth in *C. tropicalis*. In contrast, the overexpression of *DAC1* and *NAG1* did not clearly impact filamentous growth under any tested conditions (Figure 3).

### 2.4. The Impact of HXK1 Deletion on C. tropicalis Filamentous Growth and Cell Growth Rates

Given that *HXK1* overexpression enhanced *C. tropicalis* filamentous growth (Figure 3), we next explored its importance in this context using *HXK1* deletion cells. Consistent with our findings from the overexpression experiments, colonies of *hxk1*Δ/Δ mutants were smooth and failed to exhibit filamentous growth on glucose-containing plates at 25 °C (Figure 4). We similarly observed reduced *hxk1*Δ/Δ mutant cell filamentation at 37 °C on glucose-containing media (Figure 4). This suggests that *HXK1* serves to positively regulate filamentation in *C. tropicalis*, besides its role in GlcNAc catabolism. As WT cells exhibited minimal filamentation on GlcNAc-containing plates, the ability of *HXK1* deletion to further suppress filamentous growth on these plates was difficult to assess accurately (Figure 4). No clear differences in colony size were exhibited by *hxk1*Δ/Δ mutant cells relative to WT cells on glucose-containing plates, but the *hxk1*Δ/Δ mutant colonies were markedly smaller than were the WT colonies on GlcNAc-containing plates, indicating that *hxk1*Δ/Δ mutant cells grow at a reduced rate on Lee’s plus GlcNAc plates.

To confirm that the deletion of *HXK1* in *C. tropicalis* resulted in suppressed filamentous growth and that the cell growth rate was reduced on GlcNAc-containing medium, a fragment containing the *HXK1* gene was transformed back into the genome and integrated into the original *HXK1* locus. We found that the percentages of filamentous cells and the cell growth rates of the *HXK1*/*HXK1*+*HXK1p*-*HXK1* complemented strains were similar to those of the WT (Figure 4).

### 2.5. Assessment of hxk1Δ/Δ dac1Δ/Δ nag1Δ/Δ Triple Mutant C. tropicalis Filamentation and Cell Growth

*HXK1*, *DAC1*, and *NAG1* are present within a single cluster of the *C. tropicalis* genome [21], allowing for the one-step deletion of all three genes. As with the mutant cells lacking *HXK1* expression, *hxk1*Δ/Δ *dac1*Δ/Δ *nag1*Δ/Δ triple deletion mutants were smooth and failed to undergo filamentation on glucose-containing plates at 25 °C (Figure 5). Relative to WT and reconstituted *hxk1*Δ/Δ *dac1*Δ/Δ *nag1*Δ/Δ+*HXK1p*-*HXK1*+*DAC1p*-*DAC1*+*NAG1p*-*NAG1* controls, these triple mutant cells exhibited a clear reduction in filamentous growth on glucose-containing medium at 37 °C (Figure 5). Interestingly, *hxk1*Δ/Δ *dac1*Δ/Δ *nag1*Δ/Δ triple mutant colonies did not differ obviously in size relative to WT and reconstituted strain *hxk1*Δ/Δ *dac1*Δ/Δ *nag1*Δ/Δ+*HXK1p*-*HXK1*+*DAC1p*-*DAC1*+*NAG1p*-*NAG1* colonies (Figure 5). In fact, previous research has shown that *C. tropicalis* can grow on Lee’s medium without GlcNAc/glucose or other sugars as a carbon source [39]. The robust filamentous growth observed on such media is similar to that observed on Lee’s glucose medium, and so we can conclude that the decreased filamentous growth of the triple mutant of *hxk1*Δ/Δ *dac1*Δ/Δ *nag1*Δ/Δ on Lee’s GlcNAc medium is primarily a result of the GlcNAc inhibition of filamentous growth in *C. tropicalis*.

### 2.6. Assessment of the Effects of DAC1 or NAG1 Deletion on C. tropicalis Filamentous Growth

We next examined the effects of deleting *DAC1* or *NAG1* on *C. tropicalis* filamentous growth on glucose- and GlcNAc-containing media at 25 °C and 37 °C. Notably, both the *dac1*Δ/Δ and *nag1*Δ/Δ mutants were unable to grow on GlcNAc-containing plates, whereas these cells exhibited normal growth on glucose-containing plates. This suggests that GlcNAc is still able to disrupt *C. tropicalis* growth in these *dac1*Δ/Δ and *nag1*Δ/Δ mutant cells (Figure 6). Indeed, *DAC1* or *NAG1* deletion did not clearly impact growth on glucose-containing media at either tested temperature (Figure 6). We were unable to test the effects of these deletions on filamentous growth on GlcNAc-containing media, given that GlcNAc inhibited the growth of these mutant cells (Figure 6).

To confirm that the deletion of *DAC1* or *NAG1* in *C. tropicalis* inhibited the cells’ growth on GlcNAc-containing media, a fragment containing the *DAC1* or *NAG1*gene was transformed back into the genome and integrated into the original *DAC1* or *NAG1* locus, correspondingly. We found that the cell growth rates of the *dac1*Δ/Δ+*DAC1p*-*DAC1* and *nag1*Δ/Δ+*NAG1p*-*NAG1* complemented strains were similar to those of the WT (Figure 6).

## 3. Discussion

The ability to undergo advantageous morphological changes in response to environmental changes is essential to the survival of many microbes. In fungi, the yeast-to-filament transition can facilitate host tissue invasion and spread through the host circulatory system, often leading to the development of fatal systemic candidiasis, making this transition of critical importance for the pathogenicity of *Candida* species [45,46]. While *C. albicans* and *C. tropicalis* are closely related fungi, GlcNAc induces opposing outcomes in these species, promoting *C. albicans* filamentation [20,47], while inhibiting this same transition in *C. tropicalis* [39]. Glucose availability is relatively limited in environments where *C. tropicalis* is normally found, whereas GlcNAc, which is a key component of chitin and of the extracellular matrix of animals, is readily present in diverse environments, including the human gastrointestinal tract [32,33]. *Candida* and other fungal species are able to utilize GlcNAc both as a source of carbon and as a signaling molecule [34,36]. However, previous research has not assessed whether GlcNAc interferes with *C. tropicalis* filamentation in a manner dependent upon its catabolism. As such, herein we examined how the GlcNAc-mediated suppression of fungal growth is linked to GlcNAc catabolism in *C. tropicalis*.

*NGT1* is a transmembrane protein vital for GlcNAc transport into cells [36,41]. In *C. tropicalis* cells in which *NGT1* had been knocked out, there was no apparent change in fungal growth on glucose-containing plates, whereas on GlcNAc-containing plates, the *NGT1* knockout cells exhibited a clear improvement in filamentous growth at 25 °C and 37 °C (Figure 1). This thus suggests that GlcNAc entry into *C. tropicalis* cells is required for the subsequent suppression of filamentous growth. Although the deletion of *NGT1* resulted in more robust filamentous growth on Lee’s plus GlcNAc medium plates, the overexpression of *NGT1* had no obvious effect on filamentous growth under any of the four tested culture conditions (Figure 1). These phenomena can be explained by the expression level of the GlcNAc-specific transporter gene *NGT1* even in the WT control strain (GH1374), which was efficiently induced when cultured on Lee’s plus GlcNAc agar plates (Figure 2), suggesting that this GlcNAc-specific transporter is sufficient to efficiently transport GlcNAc into *C. tropicalis* cells and to thereby suppress filamentous growth.

Our results revealed that GlcNAc was only able to inhibit *C. tropicalis* filamentation following its *NGT1*-mediated internalization. *Candida* species can utilize GlcNAc as a carbon source through a catabolic mechanism dependent upon *HXK1*, *DAC1*, and *NAG1* [34,43,48]. We found that the expression of *NGT1*, *HXK1*, *DAC1*, and *NAG1* were all significantly elevated in cells grown on GlcNAc-containing media relative to those grown on glucose-containing media. Previous work suggests that in *C. albicans*, *HXK1* can suppress filamentation [44]. In *C. tropicalis*, however, we observed the opposite phenotype, with *HXK1* overexpression driving enhanced filamentous growth on both glucose- and GlcNAc-containing media (Figure 3), suggesting that this protein can positively regulate filamentous growth. Consistent with this, *HXK1* deletion mutant cells exhibited markedly reduced filamentous growth on glucose-containing media at both 25 °C and 37 °C (Figure 4). This suggests that *HXK1* is thus able to enhance filamentous growth besides GlcNAc phosphorylation (Figure 7). *C. tropicalis* may synthesize its own GlcNAc, and the overexpression of *HXK1* can eliminate GlcNAc, whereas the deletion of *HXK1* cannot eliminate GlcNAc synthesized by *C. tropicalis* itself, which may repress filamentation. *HXK1* is known to physically interact with a histone deacetylase (Sir2), which controls phenotypic switching in *C. albicans* [49,50]. This phenomenon indicates that *HXK1* may phosphorylate many substrates in the context of signal transduction [50,51], and as such this positive regulatory phenotype may be linked to the enhanced phosphorylation of particular yet-to-be-identified substrates in *C. tropicalis*.

Relative to WT cells, we observed a clear reduction in the growth rate of *hxk1*Δ/Δ mutant cells on plates containing GlcNAc, which may be a result of the dramatically increased *DAC1* and *NAG1* expression observed in these cells (Figure 2), leading to the rapid breakdown of GlcNAc-6-PO_4_, whereas in *HXK1* deletion mutants, this GlcNAc-6-PO_4_ cannot be produced. As such, we generated *hxk1*/Δ/Δ *dac1*Δ/Δ *nag1*Δ/Δ triple mutant cells and assessed their relative filamentous growth. Compared to WT cells, these triple mutants exhibited no apparent differences in colony size (Figure 5), instead exhibiting filamentous growth similar to that of the *hxk1*Δ/Δ mutant under tested conditions, with GlcNAc effectively inhibiting filamentous growth for both *hxk1*Δ/Δ and *hxk1*Δ/Δ *nag1*Δ/Δ *dac1*Δ/Δ mutant cells (Figure 5 and Figure 7). Therefore, these results further supported the ability of GlcNAc to suppress filamentation through a mechanism independent of its catabolic processing.

*DAC1* and *NAG1* failed to clearly impact filamentous growth under the tested conditions (Figure 6), suggesting that these genes do not play a critical role in the regulation of *C. tropicalis* filamentous growth. This was further supported by the observation that the *hxk1*Δ/Δ *dac1*Δ/Δ *nag1*Δ/Δ and *hxk1*Δ/Δ mutant strains exhibited similar filamentous growth under the tested conditions (Figure 5). The *dac1*Δ/Δ and *nag1*Δ/Δ mutant strains were unable to grow on GlcNAc-containing plates, whereas they grew normally on glucose-containing plates (Figure 6), suggesting that GlcNAc-6-PO_4_ accumulation induces cytotoxicity in *C. tropicalis* (Figure 7). This is consistent with findings indicating that GlcNAc is able to suppress the growth of *dac1*Δ/Δ and *nag1*Δ/Δ mutant *C. albicans* strains [44]. This was also consistent with the fact that the *hxk1*Δ/Δ *dac1*Δ/Δ *nag1*Δ/Δ triple mutant cells grew at a rate similar to WT control cells. Together, these results indicate that *hxk1*Δ/Δ cells grow substantially slower on GlcNAc-containing media, suggesting that metabolic balance is essential for normal *C. tropicalis* cellular growth.

In summary, these findings indicate that *NGT1*-mediated GlcNAc transport into *C. tropicalis* cells is essential in order for GlcNAc to inhibit filamentous growth. Furthermore, *HXK1* serves to positively regulate filamentous growth and to phosphorylate GlcNAc. We further found that excessive constitutive GlcNAc-6-PO_4_ accumulation is toxic to *C. tropicalis*. Together, these results may highlight potential avenues for future therapeutic intervention.

## 4. Materials and Methods

### 4.1. Fungal Culture

Table 1 details the strains and plasmids used in the present study. *C. tropicalis* cells were plated or patched onto solid YPD medium (20 g/L glucose, 20 g/L peptone, 10 g/L Yeast extract, and 20 g/L Agar) and were routinely grown at 25 °C. YPM medium (20 g/L maltose, 20 g/L peptone, 10 g/L yeast extract) was used for the FLP-mediated excision of the caSAT1/flipper cassette [5]. Modified Lee’s medium (5 g/L NaCl, 5 g/L (NH_4_)_2_SO_4_, 2.5 g/L K_2_HPO_4_, 0.2 g/L MgSO_4_.7H_2_O, 1.3 g/L L-leucine, 1 g/L L-lysine, 0.5 g/L L-alanine, 0.5 g/L L-phenylalanine, 0.5 g/L L-pronine, 0.5 g/L L-threonine, 0.1 g/L L-methionine, 70 mg/L L-ornithine, 70 mg/L L-arginine, 70 mg/L L-histidine, 70 mg/L uridine, 1 mg/L D-biotin and 16 ug/L ZnSO_4_) containing GlcNAc (12.5 g/L) or glucose (12.5 g/L) was used to analyze filamentous growth [5]. Solid agar plates were prepared by the addition of 20 g/L agar (BD, Franklin Lakes, NJ, USA) to Lee’s plus glucose and Lee’s plus GlcNAc media. The amino acids, inorganic salts, glucose, and GlcNAc used to construct Lee’s media were from Sigma-Aldrich (St. Louis, MO, USA).

### 4.2. Plasmid Construction

The pCT3 vector for *C. tropicalis* was used to generate overexpression plasmids through a slightly modified version of previous protocols [39]. For pCT3-HXK1 overexpression plasmid generation, the ORF of *HXK1* along with a 250 bp 3′-UTR region was amplified by PCR using *HXK1*-specific primers from GH1374 genomic DNA, with *Not*I and *Kpn*I sites being present within the amplified sequence. *Not*I and *Kpn*I were then used to digest this sequence, which was inserted into the pCT3 construct downstream of the TDH3 promoter, as TDH3 is the most highly expressed housekeeping gene in *C. tropicalis*. A previous RNA-Seq analysis found that the mRNA expression level of TDH3 was dozens to hundreds of times that of *HXK1*, *DAC1*, *NAG1*, and *NGT1* on both Lee’s glucose medium and Lee’s GlcNAc medium [39]. The pCT3-NGT1, pCT3-*DAC1*, and pCT3-NAG1 plasmids were similarly generated through the amplification of *NGT1*, *DAC1*, and *NAG1* ORFs and 250 bp 3′-UTR regions from GH1374 genomic DNA using appropriate pairs of PCR primers. *Not*I and *Pst*I were used to digest the resultant nucleic acids prior to inserting them into the pCT3 plasmid downstream of the TDH3 promoter.

The pSF2A vector was used for deletion plasmid construction, using modified versions of a previously reported protocol [53]. *NGT1*, *HXK1*, *DAC1*, and *NAG1* knockout plasmids were produced via the amplification of two flanking fragments (5′-UTR and 3′-UTR) from GH1374 genomic DNA for each gene, followed by digestion with the *Apa*I/*Xho*I and *Sac*II/*Sac*I restriction enzyme pairs prior to the insertion of the pSF2A plasmid, which contained a caSAT1 selection marker.

Appendix A details all primers used for plasmid generation in this study.

### 4.3. Overexpression and Deletion Mutant Construction

*C. tropicalis* overexpression mutants were generated based on a previous protocol [39]. Briefly, Ascl was used to linearize pCT3, pCT3-NGT1, pCT3-HXK1, pCT3-DAC1, and pCT3-NAG1, after which *C. tropicalis* (GH1374) was transformed using these plasmids, followed by selection on YPD agar plates supplemented with 100 g/L nourseothricin. PCR was used to confirm proper insertion, and qRT-PCR was used to further confirm that the TDH3 promoter was able to promote *NAG1*, *DAC1*, *HXK1*, and *NGT1* overexpression.

*NGT1*, *HXK1*, *DAC1*, and *NAG1* were deleted via a slightly modified version of a previously reported protocol [54]. *Apa*I/*Sac*I were used to linearize the pSF2A-NGT1KO, pSF2A-HXK1KO, pSF2A-DAC1KO, and pSF2A-NAG1KO constructs, after which they were used to transform *C. tropicalis* (GH1374). Transformed cells then underwent selection on YPD agar plates supplemented with 100 g/L nourseothricin, and PCR was used to confirm proper insertion. To delete the second allele of these genes, a PCR fusion strategy was employed [20]. First, the flanking fragments for these individual genes were amplified from the GH1374 gDNA, as was the *caHPH* positive selection marker. Next, fusion PCR was conducted using these fragments as templates, and the strains in which the first allele of these genes had been deleted were then transformed with the PCR-amplified fragments, followed by selection on hygromycin B (100 g/L)-containing YPD agar plates. PCR was then used to confirm gene knockout. *HXK1*, *DAC1*, and *NAG1* are close to one another on the same chromosome, and no other genes are present between these genes. As such, all three genes can be deleted in one step without accidentally deleting any other genes.

All primers used in the construction of these mutants are listed in Appendix A.

### 4.4. Construction of Complemented Strains

To construct the *NGT1*/*NGT1*+*NGT1p*-*NGT1* complemented strains, a strategy was used according to a previously reported protocol [5]. We firstly constructed plasmids pSF2A+*NGT1*, pSF2A+*HXK1*, pSF2A+*DAC1*, pSF2A+*NAG1*, and pSF2A+*HXK1*+*DAC1*+*NAG1* for the transformation of corresponding mutant strains. Two fragments (one fragment containing the ORF, 5′-UTR and 3′-UTR of *NGT1*, *HXK1*, *DAC1*, or *NAG1*; the other fragment containing a corresponding 500bp 3′-UTR flanking fragment) were amplified from GH1374 genomic DNA for each gene, followed by digestion with the *Apa*I/*Xho*I and *Sac*II/*Sac*I restriction enzyme pairs prior to the insertion of the pSF2A plasmid, which contained a caSAT1 selection marker.

The *ngt1*Δ/Δ, *hxk1*Δ/Δ, *dac1*Δ/Δ, and *nag1*Δ/Δ mutants and *hxk1*Δ/Δ *dac1*Δ/Δ *nag1*Δ/Δ triple mutant were grown in YPM medium over a 48 h period for the FLP-mediated excision of the caSAT1/flipper cassette, generating the strains *NGT1*:FLP/*NGT1*:*hph*, *HXK1*:FLP/*HXK1*:*hph*, *DAC1*:FLP/*DAC1*:*hph*, *NAG1*:FLP/*NAG1*:*hph*, and *HXK1DAC1NAG1*:FLP/*HXK1DAC1NAG1*:*hph*.

*Apa*I/*Sac*I were used to linearize the pSF2A+*NGT1*, pSF2A+*HXK1*, pSF2A+*DAC1*, pSF2A+*NAG1*, and pSF2A+*HXK1*+*DAC1*+*NAG1* constructs, after which they wre used to transform the strains *NGT1*:FLP/*NGT1*:*hph*, *HXK1*:FLP/*HXK1*:*hph*, *DAC1*:FLP/*DAC1*:*hph*, *NAG1*:FLP/*NAG1*:*hph*, and *HXK1DAC1NAG1*:FLP/*HXK1DAC1NAG1*:*hph*. Transformed cells then underwent selection on YPD agar plates supplemented with 100 g/L nourseothricin, and PCR was used to confirm proper insertion. All primers used in the construction of these mutants are listed in Appendix A.

### 4.5. C. tropicalis Transformation

*C. tropicalis* strains were transformed by electroporation as described previously [51], with slight modifications. Cells from a single colony were initially inoculated in 2 mL fresh YPD medium at 30 °C for growth overnight, after which they were transferred into 100 mL fresh YPD medium and grown at 30 °C to an optical density at 600 nm (OD600) of 1.7–1.9. Cells were then harvested and washed with 20 mL ice-cold distilled water. After cells were resuspended in 10 mL of 2 × TE buffer (10 mM Tris, 1 mM EDTA, pH 8.0), 10 mL of 0.2 M lithium acetate (Sigma Aldrich Chemie, Steinheim, Germany), pH 8.0, was added. The suspension was incubated in a rotary shaker at 150 rpm for 60 min at 30 °C. A 500 μL volume of 1 M dithiothreitol was then added, and the cells were incubated for a further 30 min at 30 °C with shaking. After the addition of 40 mL of water, the cells were centrifuged, washed sequentially in 30 mL of ice-cold water (thrice) and 10 mL of ice-cold 1 M sorbitol, resuspended in 200μL of 1 M sorbitol, and kept on ice. Next, 5 μL (~1 μg) of the linear DNA fragments were then mixed with 40 μL of electrocompetent cells, and electroporation was carried out in an Equibio electroporator (0.2 cm cuvette, 1.5 kV). After electroporation, the cells were washed in 1 mL of 1 M sorbitol, resuspended in 2 mL YPD medium, and incubated in a rotary shaker at 200 rpm for 2 h at 37 °C. The cells were plated on YPD plates containing 100 μg/mL of nourseothricin or hygromycin B, and then incubated at 37 °C for 2 days.

### 4.6. Assessment of Filamentous Growth

Filamentous growth analysis was performed as described previously [39], with slight modifications. Five individual transformants for each gene were used for these assays. Cells from a single colony were first patched onto solid YPD medium plates. After culture at 25 °C for 2 days, cells were harvested and washed with sterile distilled water (ddH_2_O) and replated onto solid Lee’s plus glucose and Lee’s plus GlcNAc medium plates. Approximately 50–70 cells were plated onto each 90 mm plate, after which they were incubated at 25 °C or 37 °C for filamentous growth analysis. Colony and cellular morphology were observed and photographed after incubation for 6 days.

### 4.7. RNA Extraction

RNA was isolated based on a slightly modified version of a previous protocol [55]. Initially, *C. tropicalis* cells isolated from a single cell were grown for 2 days on solid YPD medium plates, after which they were transferred to Lee’s media agar plates supplemented with glucose or GlcNAc. After a three-day incubation at 25 °C, cells were washed thrice in cold distilled water, and then a GeneJET RNA Purification Kit was used, based on provided protocols, to extract cellular RNA. A NanoDrop 2000 instrument (Thermo Fisher Scientific, Waltham, MA, USA) was used to assess RNA concentrations, while gel electrophoresis was used to gauge RNA integrity.

### 4.8. Quantitative Reverse Real-Time PCR (qRT-PCR)

cDNA was generated using 1 μg of total RNA from each sample (1.0 μg) with a RevertAid first-strand cDNA synthesis kit (Thermo Fisher Scientific, Waltham, MA, USA), based on provided directions, after which cDNA samples were diluted in a 180 μL total volume prior to use for qRT-PCR assays.

All qRT-PCR reactions were conducted in a 20 μL volume containing 10 μL SYBR Green Realtime PCR Master Mix (Toyobo Co., Ltd., Osaka, Japan), 2 μL diluted cDNA, 0.8 μL (10μM) of each primer (Tianyi Huiyuan Company, Wuhan, China), and 6.4 μL ddH_2_O. Reactions were run on an ABI 7500 Real-Time PCR detection system (Applied Biosystems, Foster City, CA, USA) with the following parameters: 95 °C for 60s; 40 cycles of 95 °C for 15s, 58 °C for 15 s, and 72 °C for 45 s; followed by a melting curve (65 °C to 95 °C) to confirm product specificity. For normalization, *C. tropicalis ACT1* was used as a control.

## Figures and Tables

**Figure 1 ijms-21-04036-f001:**
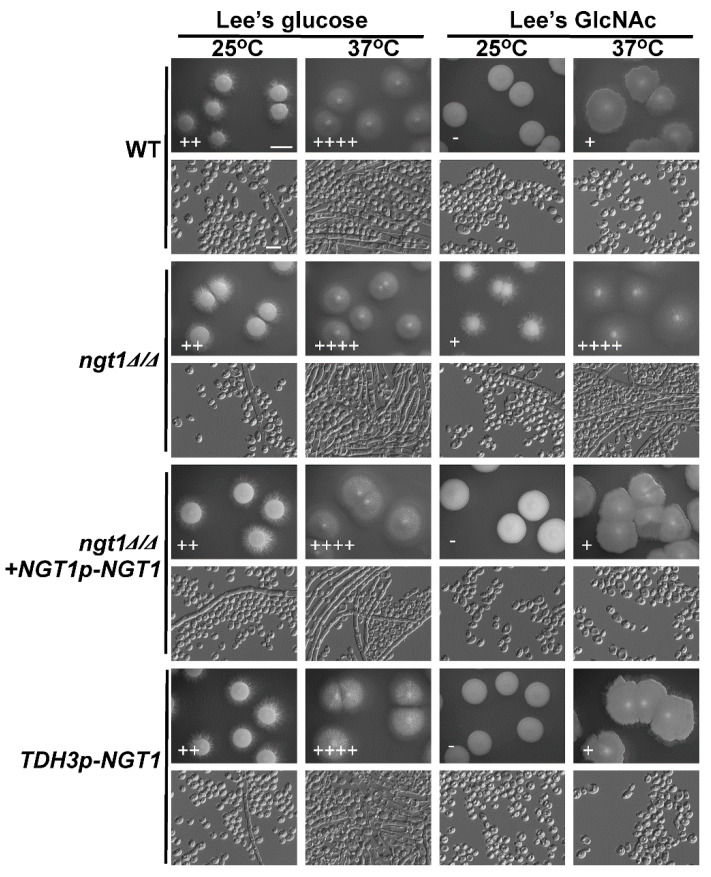
The impact of *NGT*1 deletion or overexpression on *C. tropicalis* filamentous growth. WT, *ngt1*Δ/Δ mutant, *ngt1*Δ/Δ*+NGT1p-NGT1* reconstituted strain, and *NGT1*-overexpressing mutant cells were plated on agar plates containing Lee’s media containing either glucose or *N*-acetylglucosamine (GlcNAc), and morphology was assessed after 6 days of growth at either 25 °C or 37 °C. Five individual transformants per gene were used for filamentous growth assays. The degree of hyphal growth is represented by “+” signs, with the percentages of filamentous cells being indicated as follows: +, 0.1%–5%; ++, 5%–20%; +++, 20%–40%; ++++, 40%–60%. If no hyphal growth was evident, ‘-’ was instead used to indicate cells (<0.1%). Scale bars for colonies and cells were 2 mm and 10 μm, respectively.

**Figure 2 ijms-21-04036-f002:**
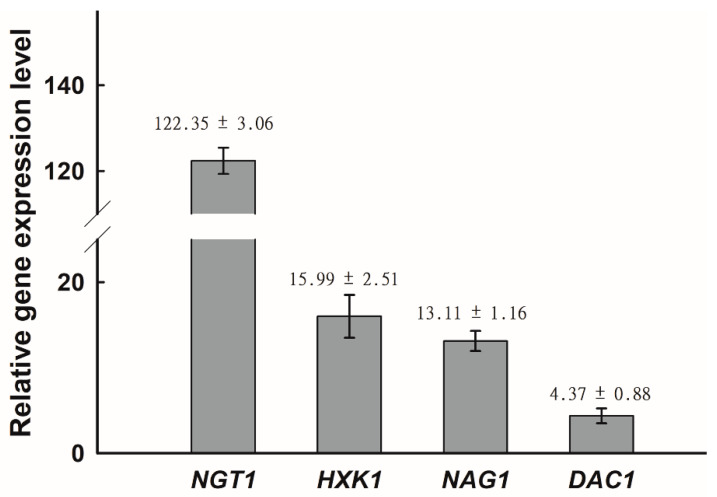
Relative expression of *NGT1*, *HXK1*, *NAG1*, and *DAC1* was significantly enhanced on GlcNAc-containing media relative to glucose-containing media. The upregulation of *NGT1*, *HXK1*, *NAG1*, and *DAC1* upon growth on Lee’s medium with GlcNAc, as compared to Lee’s medium with glucose at 25 °C, was assessed via qRT-PCR, with *C. tropicalis* actin (ACT1) being used to normalize gene expression. Gene expression was further normalized, such that expression levels on Lee’s glucose agar plates were set to 1. The average fold change ± SD is shown above the bar for each gene. Data are means ± SD from five independent biological replicates.

**Figure 3 ijms-21-04036-f003:**
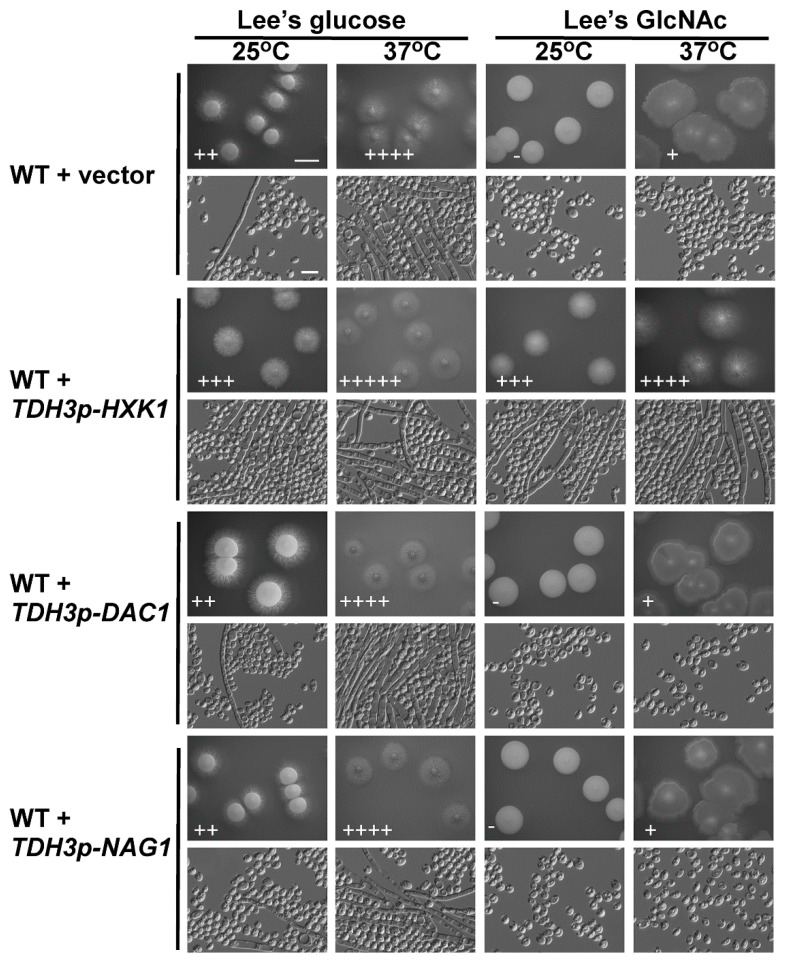
*HXK1* overexpression enhanced *C. tropicalis* filamentous growth. WT or overexpression strains were plated on Lee’s agar plates containing glucose or GlcNAc, and cellular/colony morphology was imaged after 6 days of growth at 25 °C or 37 °C. The WT + vector strain (GH1374 + pCT3) served as a control. Five individual transformants for each gene were used for filamentous growth assays. The degree of hyphal growth is represented by “+” signs, with the percentages of filamentous cells being indicated as follows: +, 0.1%–5%; ++, 5%–20%; +++, 20%–40%; ++++, 40%–60%; +++++, >60%. If no hyphal growth was evident, “-” was instead used to indicate cells (<0.1%). Scale bars for colonies and cells were 2 mm and 10 μm, respectively.

**Figure 4 ijms-21-04036-f004:**
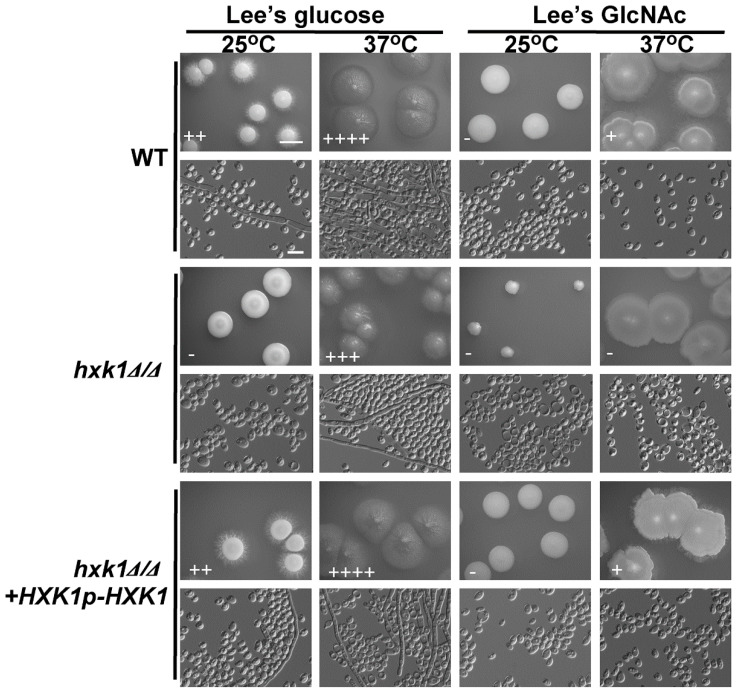
*HXK1* deletion markedly reduces *C. tropicalis* filamentous growth. Cells of WT, *hxk1*Δ/Δ mutant, and *hxk1*Δ/Δ+*HXK1p*-*HXK1* reconstituted strain were plated on Lee’s agar plates containing glucose or GlcNAc, and after 6 days at 25 °C or 37 °C, representative colony and cellular morphology was assessed. Five individual transformants for each gene were used for filamentous growth assays. The degree of hyphal growth is represented by “+” signs, with the percentages of filamentous cells being indicated as follows: +, 0.1%–5%; ++, 5%–20%; +++, 20%–40%; ++++, 40%–60%. If no hyphal growth was evident, “-” was instead used to indicate cells (<0.1%). Scale bars for colonies and cells were 2 mm and 10 μm, respectively.

**Figure 5 ijms-21-04036-f005:**
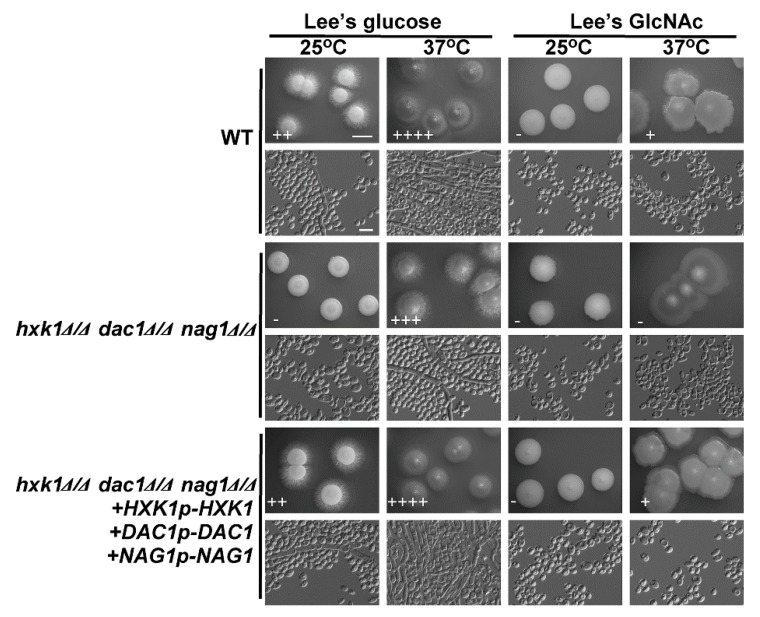
*HXK1*, *DAC1*, and *NAG1* deletion led to a marked drop in *C. tropicalis* filamentous growth. WT, *hxk1*Δ/Δ *dac1*Δ/Δ *nag1*Δ/Δ triple deletion mutant, and *hxk1*Δ/Δ *dac1*Δ/Δ *nag1*Δ/Δ+*HXK1p-HXK1*+*DAC1p*-*DAC1*+*NAG1p*-*NAG1* reconstituted strain cells were plated on Lee’s agar plates containing glucose or GlcNAc, and after 6 days at 25 °C or 37 °C representative colony and cellular morphology was assessed. Five individual transformants for each gene were used for filamentous growth assays. The degree of hyphal growth is represented by “+” signs, with the percentages of filamentous cells being indicated as follows: +, 0.1%–5%; ++, 5%–20%; +++, 20%–40%; ++++, 40%–60%. If no hyphal growth was evident, “-” was instead used to indicate cells (<0.1%). Scale bars for colonies and cells were 2 mm and 10 μm, respectively.

**Figure 6 ijms-21-04036-f006:**
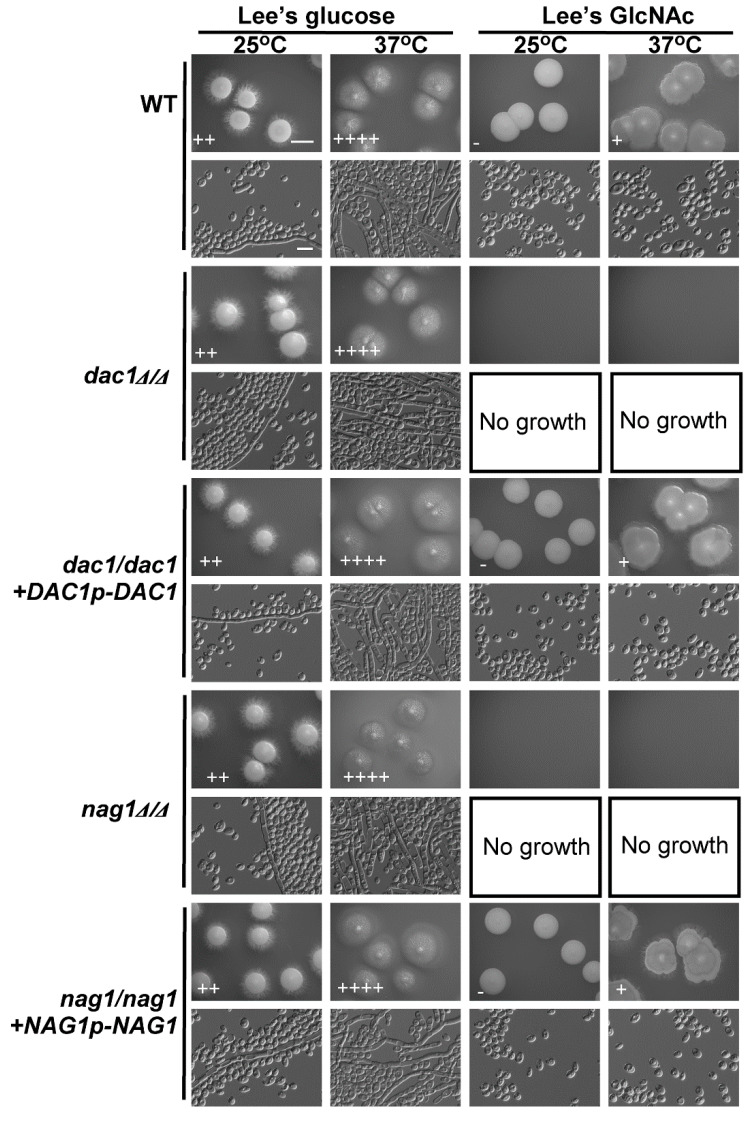
GlcNAc suppresses *dac1*Δ/Δ and *nag1*Δ/Δ cellular growth. WT, *dac1*Δ/Δ mutant, *dac1*Δ/Δ +*DAC1p*-*DAC1* reconstituted strain, *nag1*Δ/Δ mutant, and *nag1*Δ/Δ+*NAG1p-NAG1* reconstituted strain cells were plated on Lee’s agar plates containing glucose or GlcNAc, and after 6 days at 25 °C or 37 °C representative colony and cellular morphology was assessed. Five individual transformants for each gene were used for filamentous growth assays. The degree of hyphal growth is represented by “+” signs, with the percentages of filamentous cells being indicated as follows: +, 0.1%–5%; ++, 5%–20%; +++, 20%–40%; ++++, 40%–60%. If no hyphal growth was evident, “-” was instead used to indicate cells (<0.1%). Scale bars for colonies and cells were 2 mm and 10 μm, respectively. Scale bars for colonies and cells were 2 mm and 10 μm, respectively.

**Figure 7 ijms-21-04036-f007:**
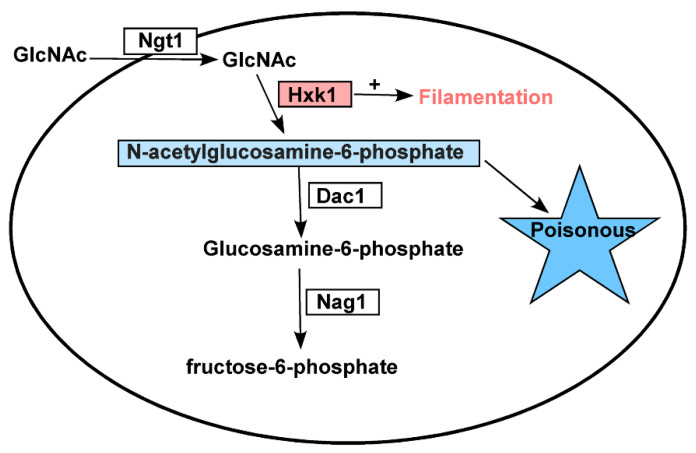
A model of how *N*-acetylglucosamine metabolism-related genes govern *C. tropicalis* functionality. *HXK1* positively regulates filamentous growth in addition to catalyzing the phosphorylation of GlcNAc, with excess GlcNAc-6-PO_4_ being toxic to *C. tropicalis*.

**Table 1 ijms-21-04036-t001:** Strains and plasmids in the present study.

Strain or Plasmid	Description	References
**Strains**		
GH1374	Natural isolate	[39]
*TDH3p-NGT1*	GH1374+pCT3-HGC1	This study
*TDH3p-HXK1*	GH1374+pCT3-HXK1	This study
*TDH3p-DAC1*	GH1374+pCT3-DAC1	This study
*TDH3p-NAG1*	GH1374+pCT3-NAG1	This study
*ngt1*Δ/Δ	As GH1374, but *ngt1:SAT1/ngt1:hph*	This study
*hxk1*Δ/Δ	As GH1374, but *hxk1:SAT1/hxk1:hph*	This study
*dac1*Δ/Δ	As GH1374, but *dac1:SAT1/dac1:hph*	This study
*nag1*Δ/Δ	As GH1374, but *nag1:SAT1/nag1:hph*	This study
*hxk1*Δ/Δ *dac1*Δ/Δ *nag1*Δ/Δ	As GH1374, but *hxk1dac1nag1:SAT1/hxk1dac1nag1:hph*	This study
*ngt1*Δ/Δ+*NGT1p-NGT1*	As GH1374, but *ngt1:FLP/ngt1:hph*+*NGT1p-NGT1:SAT1*	This study
*hxk1*Δ/Δ+*HXK1p-HXK1*	As GH1374, but *hxk1:FLP/hxk1:hph*+*HXK1p-HXK1:SAT1*	This study
*dac1*Δ/Δ+*DAC1p-DAC1*	As GH1374, but *dac1:FLP/dac1:hph*+*DAC1p-DAC1:SAT1*	This study
*nag1*Δ/Δ+*NAG1p-NAG1*	As GH1374, but *nag1:FLP/nag1:hph*+*NAG1p-NAG1:SAT1*	This study
*hxk1*Δ/Δ *dac1*Δ/Δ *nag1*Δ/Δ+*HXK1p-HXK1*+*DAC1p-DAC1*+*NAG1p-NAG1*	As GH1374, but *hxk1:FLP/hxk1:hph*+*HXK1p-HXK1:SAT1*	This study
**Plasmids**		
pCT3	*C. tropicalis* expression vector; Amp	[39]
pCT3-NGT1	*C. tropicalis NGT1* expression vector; Amp	This study
pCT3-HXK1	*C. tropicalis HXK1* expression vector; Amp	This study
pCT3-DAC1	*C. tropicalis DAC1* expression vector; Amp	This study
pCT3-NAG1	*C. tropicalis NAG1* expression vector; Amp	This study
pSF2A	Gene disruption vector containing a *caSAT1* selection marker; Chlo	[52]
pSF2A-NGT1KO	pSF2A carrying a *NGT1* knockout cassette; Chlo	This study
pSF2A-HXK1KO	pSF2A carrying a *HXK* knockout cassette; Chlo	This study
pSF2A-DAC1KO	pSF2A carrying a *DAC1* knockout cassette; Chlo	This study
pSF2A-NAG1KO	pSF2A carrying a *NAG1* knockout cassette; Chlo	This study
pSF2A-HXK1DAC1NAG1KO	pSF2A carrying a *HXK1 DAC1 NAG1* knockout cassette; Chlo	This study
pUC19T-hph-ca	The vector containing a *caHPH* selection marker; Amp	[39]
pSF2A+NGT1	Construct the *ngt1/ngt1*+*NGT1p-NGT1* complemented strains	This study
pSF2A+HXK1	Construct the *hxk1/hxk1*+ *HXK1p-HXK1* complemented strains	This study
pSF2A+DAC1	Construct the *dac1/dac1*+*DAC1p-DAC1* complemented strains	This study
pSF2A+NAG1	Construct the *nag1/nag1*+*NAG1p-NAG1* complemented strains	This study
pSF2A+HXK1+DAC1+NAG1	Construct the *hxk1*Δ/Δ *dac1*Δ/Δ *nag1*Δ/Δ+*HXK1p-HXK1*+*DAC1p-DAC1*+*NAG1p-NAG1*complemented strains	This study

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
