# Peer review of "NGT1 Is Essential for N-Acetylglucosamine-Mediated Filamentous Growth Inhibition and HXK1 Functions as a Positive Regulator of Filamentous Growth in Candida tropicalis"

_ijms, 2020, doi:10.3390/ijms21114036_

Round 1

Reviewer 1 Report

This study describes the role of NGT1 and HXK1 genes in filamentation in C. tropicalis. The experiments are very well designed in a logical fashion with appropriate controls and much appreciated. The findings reported here will have an impact on the field for developing alternative therapeutics to combat C. tropicals infections. There are few minor concerns which need to be addressed to make this study more impactful.

1. Figure 1: The picture in the panel showing filamentous growth of ngt1 mutant grown at 37C on Lee’s GlcNAc media forms the critical component of this study. Unfortunately, the filamentous growth is not very clear in this figure. If possible, the quality of this figure should be improved.

2. Methods section: Different methodologies for transformation of Candida species are available, but it is not clear which protocol is followed for this study.  A brief stepwise protocol employed in this study should be provided in the methods section. This will enhance the reproducibility of the findings.

3. The conclusions section should be at the end of discussion, not after the methods section.

4. Figure 2: It is hard to assess the fold change for DAC1. It would be better if the y-axis is either split into two fragments or if the average fold change+/-SD numbers are shown above the bar for each gene.

Minor

Line 68: Italicize "Histoplasma capsulatum"

Line 78: Space should be present between words "catabolic" and "genes"

Line 93: Change "were undergo" to "had undergone"

Line 155: Remove the word "and after WT

Line 173: Italicize "C. tropicalis"

Author Response

Point 1: Figure 1: The picture in the panel showing filamentous growth of ngt1Δ/Δ mutant mutant grown at 37C on Lee’s GlcNAc media forms the critical component of this study. Unfortunately, the filamentous growth is not very clear in this figure. If possible, the quality of this figure should be improved.

Response 1: Per your suggestion, we have improved the quality of Figure 1 and have provided an image to show the growth status of mycelia more clearly.

Point 2: Methods section: Different methodologies for transformation of Candida species are available, but it is not clear which protocol is followed for this study. A brief stepwise protocol employed in this study should be provided in the methods section. This will enhance the reproducibility of the findings.

Response 2: Per your suggestion, we have provided the C. tropicalis transformation protocol used in this study in the methods section of our revised manuscript (Lines 385-401).

Point 3: The conclusions section should be at the end of discussion, not after the methods section.

Response 3: Per your suggestion, we have moved the conclusions section after the discussion section.

Point 4: Figure 2: It is hard to assess the fold change for DAC1. It would be better if the y-axis is either split into two fragments or if the average fold change+/-SD numbers are shown above the bar for each gene.

Response 4: Figure 2: Per your suggestion, y-axis was split into two segments and we also have shown the average fold change ± SD above the bar for each gene in the revised manuscript.

Point minor comments:

Line 68: Italicize "Histoplasma capsulatum"

Line 78: Space should be present between words "catabolic" and "genes"

Line 93: Change "were undergo" to "had undergone"

Line 155: Remove the word "and" after WT

Line 173: Italicize "C. tropicalis"

Response minor comments:

Line 68: We apologize for this error; we have italicized "Histoplasma capsulatum" in the revised manuscript (Line 66).

Line 78: We apologize for this error; we have inserted a space between " catabolic " and " genes " in the revised manuscript (Line 76).

Line 93: We have changed "were undergo" to "underwent" in the revised manuscript (Line 91).

Line 155: Per your suggestion, we have removed the word "and" after WT in the revised manuscript (Line 168).

Line 173: We apologize for this error; we have italicized " C. tropicalis " in the revised manuscript (Line 177).

Reviewer 2 Report

This is a study of the role of GlcNAc in filamentation of C. tropicalis. While both C. albicans and C. tropicalis filament, in C. albicans filamentation is activated by GlcNAc and in C. tropicalis it is repressed by GlcNAc. The authors demonstrate that in mutants of the NGT1 uptake transporter of GlcNAc that C. tropicalis can maintain filamentation in the presence of GlcNAc. They go on to demonstrate that a suite of GlcNAc catabolic genes are upregulated by the presence of GlcNAc and that overexpression of the first gene in that pathway (HXK1) results in hyperfilamentation while increased expression of the others does not affect filamentation. Loss of HXK1 also results in decreased filamentation. A triple mutant of HXK1, DAC1 and NAG1 phenocopies the HXK1 mutant for the filamentation defect. Interestingly, DAC1 and NAG1 mutants are unable to grow on GlcNAc media, although the authors note this has previously been observed in C. albicans.

The data are relatively clean and the authors are admirably thorough with their use of complementation controls. However, I do have a couple questions about the conclusions.

First- the authors conclude that the HXK1 overexpression phenotype means that "The ability of HXK1 to enhance filamentous growth was not linked to its ability to drive GlcNAc catabolism, given that it offered benefits on medium containing both glucose and GlcNAc (Figure 3)." This is only a valid conclusion if there is no other possible source of GlcNAc aside from that provided exogenously in the media. Does C. tropicalis synthesize its own GlcNAc? Even if this occurs at very low levels, overexpression of HXK1 could eliminate this pool and alleviate low level repression of filamentation by GlcNAc. This model would also be consistent with the results observed by overexpression of the downstream parts of the GlcNAc catabolic pathway. Given that this is a component of chitin and UDP-GlcNAc is fairly ubiquitous in fungi, it seems likely that C. tropicalis synthesizes its own as well, although a brief literature search didn't turn anything up.

Second- If GlcNAc is the only carbon source in the Lee's + GlcNAc media, how does a triple mutant of hxk1, dac1, and nag1 grow on this medium? There must be other catabolic genes/pathways not described or assessed here. Perhaps filamentation is being affected by one of these other pathways.

Minor comments:

Figure 2 is a little unclear. It appears that this is showing the upregulation of these genes upon growth on Lee's media at 25 with GlcNAc as compared to Lee's media at 25 with glucose. But the figure legend could say this more clearly.

The recipe for Lee's medium is not provided in methods.

The title of Table 1 appears to be incorrect.

Line 68: Histoplasma capsulatum should be in italics

Line 73: "There is few research" should be "There is little research"

Line 78: Missing space in 'catabolicgenes"

Line 93: "were undergo" should be "underwent"

Author Response

Point 1: First- the authors conclude that the HXK1 overexpression phenotype means that "The ability of HXK1 to enhance filamentous growth was not linked to its ability to drive GlcNAc catabolism, given that it offered benefits on medium containing both glucose and GlcNAc (Figure 3)." This is only a valid conclusion if there is no other possible source of GlcNAc aside from that provided exogenously in the media. Does C. tropicalis synthesize its own GlcNAc? Even if this occurs at very low levels, overexpression of HXK1 could eliminate this pool and alleviate low level repression of filamentation by GlcNAc. This model would also be consistent with the results observed by overexpression of the downstream parts of the GlcNAc catabolic pathway. Given that this is a component of chitin and UDP-GlcNAc is fairly ubiquitous in fungi, it seems likely that C. tropicalis synthesizes its own as well, although a brief literature search didn't turn anything up.

Response 1: We agree that. C. tropicalis may synthesizes its own GlcNAc, and that the overexpression of HXK1 could eliminate the GlcNAc synthesized by C. tropicalis itself, thereby potentially repressing filamentation. We have revised the conclusions of the revised manuscript to reflect this possibility (Lines 144-147, 272-274).

Point 2: Second- If GlcNAc is the only carbon source in the Lee's + GlcNAc media, how does a triple mutant of hxk1Δ/Δ dac1Δ/Δ nag1Δ/Δ grow on this medium? There must be other catabolic genes/pathways not described or assessed here. Perhaps filamentation is being affected by one of these other pathways.

Response 2: In our present study (data not shown) and a previous study (Zhang et al 2016), it was shown that C. tropicalis can grow on Lee's medium without GlcNAc/glucose or other sugars as a carbon source, and that the robust filamentous growth observed on such media is similar to that observed on Lee's glucose medium. As such, we can conclude that the decreased filamentous growth of the triple mutant of hxk1Δ/Δ dac1Δ/Δ nag1Δ/Δ on Lee's GlcNAc medium is mainly the results of GlcNAc inhibition of filamentous growth in C. tropicalis. We have discussed this phenomenon in the revised manuscript (Lines 195-200).

Point 3: Figure 2 is a little unclear. It appears that this is showing the upregulation of these genes upon growth on Lee's media at 25°C with GlcNAc as compared to Lee's media at 25°C with glucose. But the figure legend could say this more clearly (Minor comments).

Response 3: Per your suggestion, we have revised the legend for Figure 2 to emphasize the upregulation of NGT1, HXK1, NAG1, and DAC1 upon growth on Lee's media with GlcNAc as compared to Lee's media with glucose at 25°C in the revised manuscript (Lines 120-123).

Point 4: The recipe for Lee's medium is not provided in methods (Minor comments).

Response 4: Per your suggestion, we have provided the recipe for Lee's medium in methods in the revised manuscript (Lines 317-320).

Point 5: The title of Table 1 appears to be incorrect. (Minor comments).

Response 5: We apologize for this mistake in the title of Table 1. We have corrected it in the revised manuscript (Line 324).

Point 6: Line 68: Histoplasma capsulatum should be in italics

Response 6: We apologize for this error; we have italicized "Histoplasma capsulatum" in the revised manuscript (Line 66).

Point 7: Line 73: "There is few research" should be "There is little research" (Minor comments).

Response 7: Per your suggestion, we have changed "there is few research " to "there is little research" in the revised manuscript (Line 71).

Point 8: Line 78: Missing space in 'catabolicgenes"

Response 8: We apologize for this error; we have inserted a space between " catabolic " and " genes " in the revised manuscript (Line 76).

Point 9: Line 93: "were undergo" should be "underwent" (Minor comments).

Response 9: Per your suggestion, we have changed "were undergo" to "underwent" in the revised manuscript (Line 91).

Round 2

Reviewer 2 Report

I'm satisfied with the responses from the authors, primarily via providing caveats for their previous conclusion. I believe the new version is ready to publish.

I did spot a couple of new typos (below):

Revised line 331: Pronine should be proline

332: urdine should be uridine